# The Characterization of Fatigue Damage of 316L Stainless Steel Parts Formed by Selective Laser Melting with Harmonic Generation Technique

**DOI:** 10.3390/ma15030718

**Published:** 2022-01-18

**Authors:** Rui Qiao, Xiaoling Yan

**Affiliations:** School of Artificial Intelligence, Beijing Technology and Business University, Beijing 102488, China; qiaorui2121@163.com

**Keywords:** selective laser melting, fatigue damage, dislocation, crack, nonlinear ultrasonic

## Abstract

Fatigue damage is the main reason for the failure of parts formed by selective laser melting (SLM) technology. This paper presents a high-order, harmonic, and nonlinear ultrasonic testing system for monitoring the generation and evolution of fatigue damage in SLM 316L stainless steel parts. The results demonstrate that the normalized ultrasonic, nonlinear coefficients show a significant dependence on the degree of fatigue damage of the tested specimen and that the normalized, ultrasonic, and nonlinear coefficients are effective in characterizing the degree of fatigue damage in SLM 316L stainless steel parts. Transmission electron microscope (TEM) and scanning electron microscope (SEM) analyses show that the variation in the normalized, ultrasonic, nonlinear coefficients reflect the generation and evolution process of dislocation and crack in the fatigue process of SLM 316L stainless steel specimens, and reveal the fatigue damage mechanism of SLM 316L stainless steel parts.

## 1. Introduction

Selective laser melting technology has been used extensively in a number of fields, including the aerospace, biomedicine, and automobile industries [1,2,3,4]. However, there are some micro defects (pores and cracks) in SLM 316L stainless steel parts [5,6] that affect the bearing capacity of the part under fatigue loading. Traditional NDT technologies (such as X-ray [7,8], ultrasound [9,10,11], eddy current [12,13], penetrant [14,15], and magnetic powder [16]) have limitations in the detection of micro defects. Classical nonlinear acoustics theory shows that second harmonics are mainly induced by a harmonicity of the crystal lattice of materials [17,18,19]. The nonclassical nonlinearity effect is caused by micro defects (cracks and pores) [20,21,22,23]. This paper presents a nonlinear ultrasonic testing system for monitoring the generation and evolution of fatigue damage in SLM 316L stainless steel parts.

Nonlinear ultrasonic detection methods can be divided into the high-order harmonics detection method [24,25], the mixing modulation method [26,27,28], and the ultrasonic phased array imaging detection method [29], of which the high-order harmonics detection method is the most mature and widely used. Addressing the problem that SLM 316L stainless steel parts are prone to fatigue damage in multiple working conditions, and based on the nonlinear acoustics theory, in this study, the nonlinear ultrasonic test for the SLM 316L stainless steel specimen, which was subjected to fatigue loading, was carried out using the high-order harmonics detection method. The research results show that the ultrasonic nonlinear coefficients are sensitive to the fatigue damage of the SLM 316L stainless steel specimen. Combining the results of the nonlinear ultrasonic testing with SEM and TEM analysis, the fatigue damage mechanism of 316L stainless steel parts formed by selective laser melting are analyzed in this paper.

## 2. Generation of High-Order Harmonics

When a one-dimensional longitudinal wave propagates in solid media, the nonlinear effect caused by the change in the microstructure and the presence of microcracks in the material can be described by the nonlinear wave equation. The nonlinear wave equation was established by Cantrell [30].

In one-dimensional cases, the motion of the nonlinear wave equation can be expressed as:(1)$$\rho \frac{\partial^{2}u}{\partial t^{2}}=\frac{\partial \sigma }{\partial x}$$
where *ρ* is the density of the medium, *u* is the displacement in the *x* direction, *σ* (*x*,*t*) is the normal stress in the *x* direction, and *σ* is time-dependent.

When the deformation in a solid medium is very small, the normal strain *ε*(*x*,*t*) can be defined as:(2)$$\varepsilon =\frac{\partial u}{\partial x}$$

According to the nonlinear principle of solid media, the stress–strain relation is:(3)$$\sigma =E\varepsilon (1-\frac{1}{2}\beta \varepsilon +\frac{1}{3}\delta \varepsilon^{2}\ldots )$$
where *E* is the elastic modulus and *β* and *δ* are the nonlinear coefficients related to the material. Simultaneous Equations (1)–(3) ignore the higher-order terms (higher than the second-order) in the equation, and the resultant nonlinear wave equation can be expressed as:(4)$$\;\frac{\partial^{2}u}{\partial t^{2}}=c\left[1-\beta \left(\frac{\partial u}{\partial x}\right)\right]\frac{\partial^{2}u}{\partial x^{2}}$$
where *t* is the propagation time (s); *c* is the ultrasonic wave propagation velocity (m/s); *x* is the propagation distance (m). The initial conditions are set as:(5)$$u\left(0,t\right)=A_{1}\sin\left(\omega t\right)$$

Then, according to the principle of the perturbation method, the second-order approximate solution of wave Equation (4) can be written as:(6)$$u\left(x,t\right)=A_{1}\sin\left(kx-\omega t\right)+\frac{1}{8}\left(A_{1}{}^{2}k^{2}\beta x\right)\cdot \cos2\left(kx-\omega t\right)$$
where *A*_1_ is the amplitude of the fundamental wave, *A*_2_ is the amplitude of the second harmonics, *k* = *w*/*c* is the wave number, *ω* is the frequency, *β* is the second-order nonlinear coefficient, and the expression of *A*_2_ is:(7)$$A_{2}=\frac{1}{8}\left(A_{1}{}^{2}k^{2}\beta x\right)$$

According to Equation (7), the expression of *β* is:(8)$$\beta =8\left(\frac{A_{2}}{A_{1}{}^{2}}\right)\frac{1}{k^{2}x}$$

The propagation distance and the wave number are usually constant in the process of ultrasonic testing. Thus, the express of *β* can be written as:(9)$$\beta =\frac{A_{2}}{A_{1}{}^{2}}$$

The value of *β* can be calculated by measuring *A*_1_ and *A*_2_, and *β* can be used to characterize the damage degree of the tested parts.

## 3. Experiment Procedure

### 3.1. Specimen Preparation

The specimens were prepared using 316L stainless steel spherical powder and SLM technology. The maximum particle size of the powder was 60 μm and the minimum particle size was 45 μm, the apparent density was 4.42 g/cm^3^. The chemical composition of the powder is shown in Table 1.

The specimens were prepared using an AM400 (Renishaw plc, Gloucestershire, UK) additive manufacturing system. The main processing parameters are shown in Table 2. The build direction was vertical. Three groups of specimens (group A, B, and C) were prepared. The tension tests were conducted for group A specimens (*A*1 to *A*3), and the high-cycle fatigue loading tests were conducted for group B specimens (B1 to B3). The high-cycle fatigue loading tests and the TEM and SEM tests were conducted for group C specimens (C1 to C6). The size of the specimen is shown in Figure 1, and the size is in millimeters (mm).

### 3.2. Experimental System and Measurement Method

The tension tests for the SLM 316L stainless steel specimens (*A*1 to *A*3) were carried out. According to the experimental results, the mean yield strength of the specimens was 503 MPa and the mean maximum strain was 29.7%. The high-cycle fatigue loading tests for the SLM 316L stainless steel specimens (B1 to B3) were carried out on an Instron 8801 (Instron, Boston, MA, USA) fatigue testing machine. The fatigue load is shown in Figure 2. The maximum loading stress was 400 MPa (400 MPa < 503 MPa), the stress ratio was 0.1, and the fatigue loading frequency was 10 Hz.

As shown in Figure 3, the nonlinear ultrasonic experimental system consisted mainly of an RITEC RAM-5000 SNAP (RITEC, Rochester, NY, USA) nonlinear, high-energy ultrasonic testing system, transmitting and receiving transducers (SIUI, Shantou, China), TPO40000 oscilloscope (Tektronix, Beaverton, OR, USA), matching resistance, and a computer.

When the specimen was subjected to fatigue loading, and the fatigue cycles reached the set number, the fatigue loading experimental system stopped (once every 5000 fatigue cycles), and the tensile load remained unchanged. Prior to applying the TM-100 medical ultrasonic couplant (Xiyuan temple, Tianjin, China) to the transducers and the surface of the specimen, the surface of the specimen was wiped to keep the contact surface completely clean. Then, two transducers were mounted separately on the surface of the specimen for ultrasonic transmitting and receiving, and the fixed positions of the transducers are shown in Figure 3. The specimen and the transducers were winded with an elastic band, to ensure the fixed positions of the transducers were stable during the experiment. High-frequency ultrasonic provided a better resolution of the damage to the tested specimen. In this experiment, the generation frequency (*f*_0_) of the trigger signal was chosen as 5 MHz, and the receiving signal was set to 10 MHz. The receiving transducer used in this experiment had a good sensitivity at 2*f*_0_. When fatigue damage was generated in the tested specimen, the second harmonics were generated when the ultrasonic wave passed through the corresponding area. The process was repeated until the test specimen broke, whereupon the experiment was stopped.

## 4. Experimental Results

### 4.1. Nonlinearity Coefficients in the SLM 316L Stainless Steel Specimen

The nonlinear ultrasonic test for specimen B1 (as shown in Figure 4) was carried out before the fatigue loading test. The time-domain signal received by the receiving transducer is shown in Figure 5a, the frequency spectra of the received signal are shown in Figure 5b. The high-cycle fatigue loading test for specimens B1 was carried out, and when the fatigue cycles reached 10,000 and 50,000, the time-domain signal received by the receiving transducer was recorded and is shown in Figure 6a,c. The corresponding frequency spectra are shown in Figure 6b,d. The experimental results indicate that although the fatigue damage degree of the tested specimen was quite different in three cases, we observed similar detection results in Figure 5a and Figure 6a,c. Therefore, it was very difficult to use the received time-domain signal to characterize the fatigue damage degree of the tested specimen. As shown in Figure 5b and Figure 6b,d, it can be observed that with the increase in fatigue cycles, *A*_1_ (the amplitude of fundamental wave) decreased and *A*_2_ (the amplitude of second harmonic wave) increased significantly. Thus, nonlinear coefficients *β* = *A*_2_/*A*_1_^2^ were effective in characterizing the fatigue damage degree of the SLM 316L stainless steel specimen.

As mentioned above, when the specimen was not subjected to fatigue loading, the nonlinear ultrasonic effects appeared when ultrasonic waves passed through the detection area. The primary reason for this was that the transducers, power amplifier, and couplant were nonlinear in the experiment. Additionally, under the same experimental conditions, the nonlinear ultrasonic effect strengthens with the increase in fatigue cycles. Therefore, the accumulation of fatigue damage caused by fatigue loading is the main reason for the enhancement in the nonlinear ultrasonic effect. According to the principle of high-order harmonics, the values of *β* (the nonlinear coefficient of the specimen subjected to fatigue loading) and *β*_0_ (the nonlinear coefficient of the specimen not subjected to fatigue loading) were calculated. The normalized nonlinear coefficient *β*/*β*_0_ was used in the experiment to characterize the fatigue damage degree of the tested specimen.

The SLM 316L stainless steel specimens (B1–B3) subjected to different fatigue cycles were tested according to the experimental method described above. The results of the experiment are plotted in Figure 7. The received time-domain signals were collected three times, at each predetermined fatigue cycle. The Gaussian curve fitting method was used to process the ultrasonic test results. It can be concluded that the relationship between the normalized ultrasonic nonlinear coefficients and the fatigue cycles is roughly in a mountain-shaped curve. The error bars are plotted in each subplot, and they present the standard deviations of the normalized ultrasonic nonlinear coefficients in the SLM 316L stainless steel specimens. It can be observed that the variation range of each error bar is relatively small, indicating that the normalized ultrasonic nonlinear coefficients were a reliable means of characterizing the fatigue damage degree of the SLM 316L stainless steel specimen.

In this experiment, the relationship between the normalized ultrasonic nonlinear coefficients and the fatigue cycles was studied by analyzing three SLM 316L stainless steel specimens. The results are shown in Figure 7. Although the materials and the SLM processing parameters of the three specimens were the same, the normalized nonlinear coefficients always fluctuated at a certain level; the primary reason for this fluctuation is that the microstructure of tested specimens presented typical anisotropic characteristics [31,32,33,34]. Due to a high-energy laser beam being used as a mobile heat source during the process of the formation of the 316L stainless steel specimen using SLM technology, and through rapid heating, melting, and solidification, extreme non-equilibrium conditions were produced during the material’s processing. It was also found that the transducers, power amplifier, and coupling problems that existed in the experiment resulted in some accidental errors in the ultrasonic test results.

Due to the specimens (B1–B3) being prepared with the same materials and SLM process parameters, the ultrasonic test results in Figure 7 fluctuate only slightly, and the changing trend in the normalized ultrasonic nonlinear coefficients-fatigue cycles curve is basically the same. As can be seen from Figure 8, the normalized ultrasonic nonlinear coefficients–fatigue cycles curve can be divided into three stages. In the first stage of fatigue loading, the normalized ultrasonic nonlinear coefficients gradually increase with the increase in fatigue loading cycles, and the increase rate is slow. In the second stage of fatigue loading, the normalized ultrasonic nonlinear coefficients continue to increase to the peak at a faster rate. In the third stage of fatigue loading, the normalized ultrasonic nonlinear coefficients decrease with the increase in fatigue cycles.

As shown in Figure 8, the variation in the ultrasonic nonlinear coefficients–fatigue cycles curve reflects the generation and evolution of fatigue damage in the tested specimen during fatigue loading. The microstructure of the material affects its macro performance. Therefore, the essential reason for the change in the ultrasonic nonlinear coefficients is that the microstructure of the SLM 316L stainless steel specimens changed under different fatigue cycles. The change mechanism of the ultrasonic nonlinear coefficients was revealed in combination with the microstructure of the SLM 316L stainless steel specimen.

Group C specimens (C1 to C6) are shown in Figure 9. The specimens with different fatigue damage degrees were prepared by subjecting them to different fatigue cycles. The fatigue loading results are shown in Table 3. When the fatigue cycles reached 68,920, specimen C6 broke.

### 4.2. Fatigue Fracture Analysis

The SEM photographs of the fatigue fracture are shown in Figure 10. Figure 10a shows the fatigue source, which is distributed in a herringbone shape and is located on the surface of specimen C6, with a smooth and flat morphology. The fatigue expansion zone is shown in Figure 10b,c, and the fatigue striations can be observed in the fatigue expansion zone. The instantaneous fracture zone is shown in Figure 10d–f. The surface of the instantaneous fracture zone is uneven and fluctuated to a large extent. The surface is gray without metallic luster, and many dimples and holes are distributed on the surface, which is in line with the characteristics of a ductile fracture.

Fatigue fracture analysis indicates that the SLM 316L stainless steel specimen had evident ductile fracture properties. Therefore, the tested specimen appeared to undergo plastic deformation during the fatigue loading, where the specimen deformed dominantly by dislocation movement. In order to reveal the fatigue damage mechanism of the SLM 316L stainless steel parts, the specimens were observed using a TEM.

### 4.3. Fatigue Damage Mechanism of SLM 316L Stainless Steel Parts

The TEM photograph of specimen C1 is shown in Figure 11a, which is mainly a dislocation line and planar dislocation. Figure 11b shows the TEM photograph of specimen C2. The number of dislocation lines increased and a dislocation tangle appeared. The planar dislocation was mainly monopole distributed. Classical nonlinear acoustics theory shows that second harmonics are mainly induced by the harmonicity of the crystal lattice of materials. The theoretical research of Hikata et al. [35,36,37,38] showed that the amplitude of the second harmonics caused by dislocation is directly proportional to the fourth power of dislocation density. Therefore, in the first stage of fatigue loading, the normalized ultrasonic nonlinear coefficients tended to increase slowly, because the planar dislocation density in the microstructure of the test specimen increased with the increase in fatigue cycles. Figure 11c shows the TEM photograph of specimen C3. With the increase in fatigue cycles, the dislocation tangle further increased, and at that time, the dislocation vein gradually appeared and the dislocation vein was thicker. When the fatigue cycles reached 40,000, the fatigue damage of the test specimen was further accumulated. The TEM photograph of specimen C4 is shown in Figure 11d. It can be observed that the thicker dislocation vein gradually evolved into a dislocation wall, and the dislocation wall was a non-planar dislocation. The experimental results show that in the second stage of fatigue loading, the growth rate of the normalized nonlinear coefficients accelerated. There are two accounts that explain this phenomenon: the thicker dislocation vein gradually evolved into a more dense dislocation wall and the emergence of the microcrack. With the propagation of the microcrack, the normalized nonlinear coefficients reached the maximum. Figure 11e,f shows the TEM photographs of specimen C5 and specimen C6. A clear equiaxed dislocation cell can be seen in the two photographs, and the dislocation density tended to increase with the increase in fatigue cycles. The experimental results show that in the third stage of fatigue loading, the normalized nonlinear coefficients decreased with the increase in fatigue cycles. The reason for this lies in the finding that the microcrack gradually propagated into a macrocrack.

To further confirm that the presence of the crack in the tested specimen is reasonably accounted for by the variation in the normalized ultrasonic nonlinear coefficients in the second and third stages of fatigue loading, the test specimens were observed with a scanning electron microscope. As shown in Figure 12a,b, there was no microcrack observed in the SEM photographs, but scratches on the surfaces of specimens could be clearly observed. Therefore, in the first stage of fatigue loading, the multiplication of planar dislocation is the reason for the increase of normalized ultrasonic nonlinear coefficients. Figure 12c shows the SEM photograph of specimen C3. The microcrack appeared in the microstructure of the test specimen. When the ultrasonic wave passed through the microcrack, the high-frequency vibration of the ultrasonic wave produced periodic tensile and compressive loads on the two surfaces of the microcrack, and the two surfaces of the microcrack produced an opening and closing phenomena, which is called the breathing effect. Figure 13 is a schematic diagram of the breathing effect. Additionally, when the microcrack closed, it produced an ultrasonic nonlinear effect [39,40,41]. With the propagation of the microcrack, the closure area of the microcrack caused by the breathing effect increased, and the ultrasonic nonlinear effect was more obvious. Therefore, the normalized ultrasonic nonlinear coefficients increased gradually with the propagation of the microcrack. Figure 12d shows the SEM photograph of specimen C4. It can be observed that the microcrack propagated gradually compared to that seen in specimen C3. The relevant literature [39,40,41] shows that the nonlinear ultrasonic effect caused by microcracks is greater than that caused by dislocation, so in the second stage of fatigue loading, with the accumulation of fatigue degree, the microcrack gradually propagated. The propagation of the microcrack led to the enhancement of the breathing effect, and the growth rate of the normalized ultrasonic nonlinear coefficients accelerated and reached the maximum value.

The experimental results show that the propagation direction of the microcrack in the SLM 316L stainless steel specimen was perpendicular to the fatigue loading direction. A schematic diagram of the ultrasonic propagation direction and crack direction is shown in Figure 14. Figure 15a shows the SEM photograph of specimen C5. The microcrack propagated into a macrocrack. When the distance between the macrocrack surface was greater than the maximum displacement of the ultrasonic wave vibration, the ultrasonic wave could not directly pass through the crack, and only a small amplitude of second harmonics was generated at the macrocrack internal to the specimen. As a result, the normalized ultrasonic nonlinear coefficients gradually decreased in the third stage of fatigue loading. Figure 15b shows the macrocrack internal to the specimen fracture.

## 5. Conclusions

(1)In this study, a high-order harmonics detection method was used to explore the dependence of normalized ultrasonic nonlinear coefficients on the internal fatigue damage of SLM 316L stainless steel specimens. For this purpose, an effective testing system was established, and fatigue loading and nonlinear ultrasonic tests were carried out. TEM and SEM analyses were conducted.(2)The experiment results demonstrate that the normalized ultrasonic nonlinear coefficients *β*/*β*_0_ show a significant dependence on the number of fatigue cycles and that *β*/*β*_0_ obviously grows at a high rate once microcracks appear. The dependence of *β*/*β*_0_ on the number of fatigue cycles indicates the sensibility of this method, and *β*/*β*_0_ can effectively be used to characterize the fatigue damage degree in SLM 316L stainless steel parts caused by fatigue loading. The results of this experiment also show that the nonlinear ultrasonic effects appeared when the tested specimen was not subjected to fatigue loading. Transducers, power amplifier, and coupling problems existed in the experiment, which indicates that the inherent nonlinear effects should be considered in characterizing micro fatigue damage of SLM 316L stainless steel parts with *β*. The experimental results show that the normalized ultrasonic nonlinear coefficients are effective for characterizing the fatigue damage degree of SLM 316L stainless steel parts.(3)The normalized ultrasonic nonlinear coefficient is sensitive to the fatigue damage degree of SLM 316L stainless steel parts caused by fatigue loading. The relationship between the normalized ultrasonic nonlinear coefficients and the fatigue cycles was roughly in a mountain-shaped curve. The variation in the mountain-shaped curve can be divided into three stages. In the first stage, the multiplication of planar dislocation is the reason for the increase in the normalized ultrasonic nonlinear coefficients. In the second stage, with the accumulation of fatigue degree, the microcrack appears and gradually propagates, and the thicker dislocation vein gradually evolved into a more dense dislocation wall, so that the normalized ultrasonic nonlinear coefficients accelerate and reach the maximum value. In the third stage, due to the propagation of the microcrack into a macrocrack, the normalized ultrasonic nonlinear coefficients gradually decrease.(4)TEM and SEM analyses demonstrated that the variation in the mountain-shaped curve reflects the generation and evolution process of dislocation (such as planar dislocation, dislocation tangle, dislocation vein, dislocation wall, and dislocation cell) and cracks (microcracks, propagation of microcracks, and macrocracks) in the fatigue process of SLM 316L stainless steel parts, thereby revealing the fatigue damage mechanism of SLM 316L stainless steel parts.

## Figures and Tables

**Figure 1 materials-15-00718-f001:**
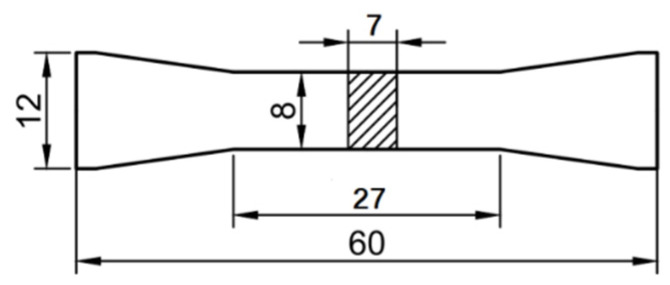
Schematic diagram of the SLM 316L stainless steel specimen.

**Figure 2 materials-15-00718-f002:**
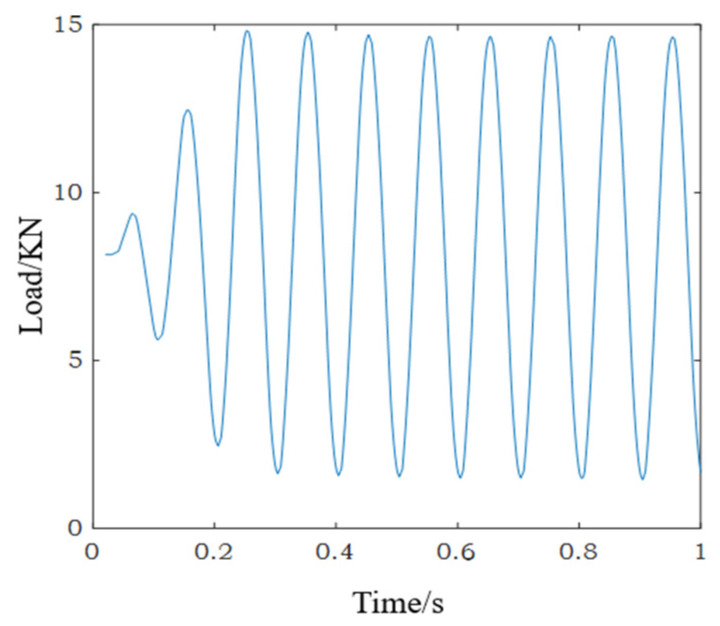
The fatigue load.

**Figure 3 materials-15-00718-f003:**
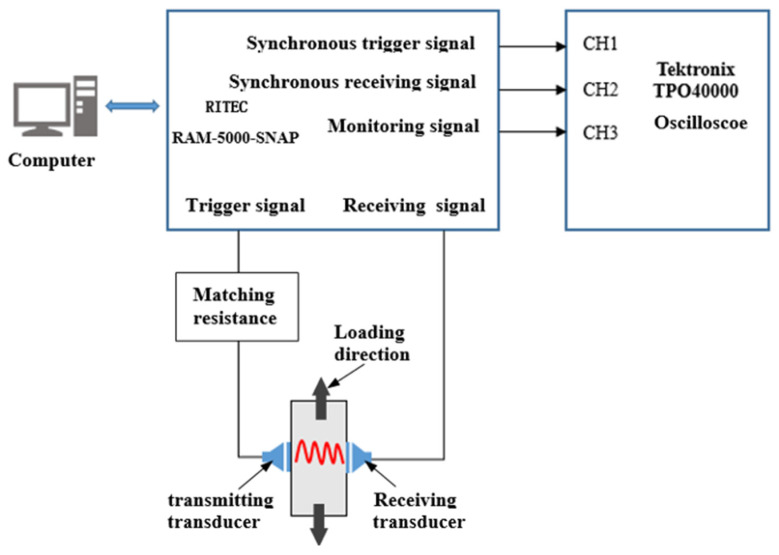
Schematic diagram of the nonlinear ultrasonic experimental system.

**Figure 4 materials-15-00718-f004:**
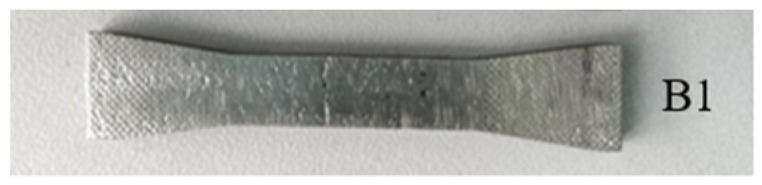
The SLM 316L stainless steel specimen.

**Figure 5 materials-15-00718-f005:**
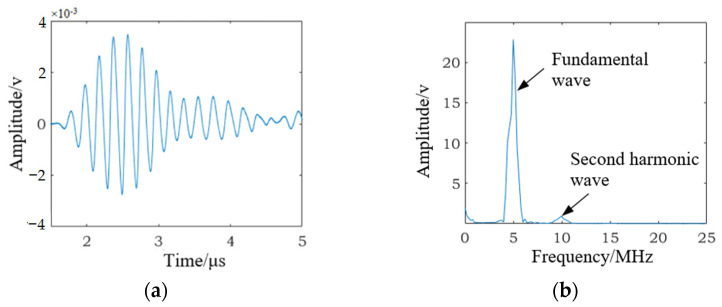
The SLM 316 L stainless steel specimen not subjected to fatigue cycles. (**a**) The received time-domain signal. (**b**) The frequency spectra of the received signal.

**Figure 6 materials-15-00718-f006:**
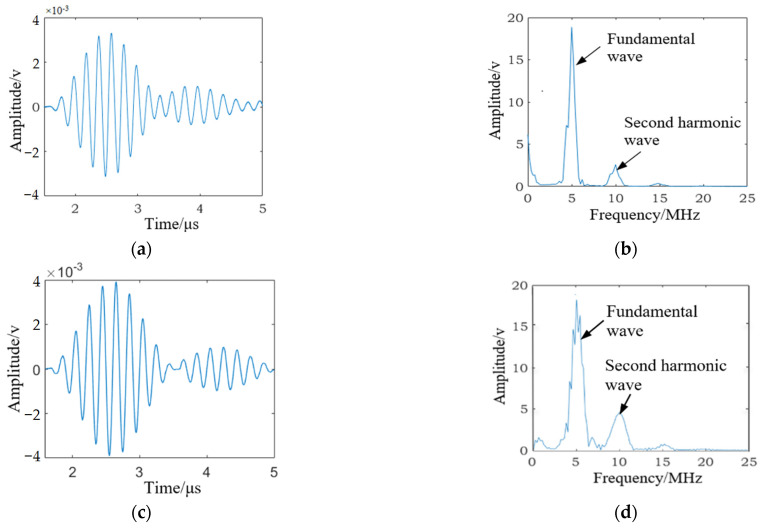
The SLM 316 L stainless steel specimen subjected to different fatigue cycles. (**a**) The received time-domain signal subjected to 10,000 fatigue cycles. (**b**) The frequency spectra of the received signal subjected to 10,000 fatigue cycles. (**c**) The received time-domain signal subjected to 10,000 fatigue cycles. (**d**) The frequency spectra of the received signal subjected to 10,000 fatigue cycles.

**Figure 7 materials-15-00718-f007:**
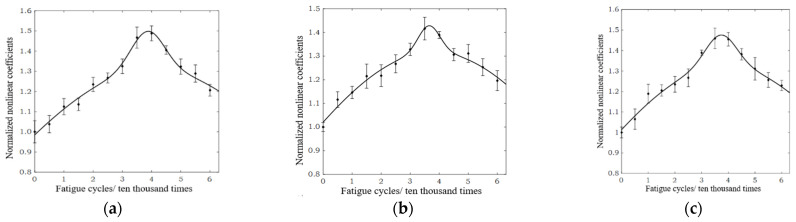
The normalized ultrasonic nonlinear coefficients vary with the fatigue cycles. (**a**) Specimen B1. (**b**) Specimen B2. (**c**) Specimen B3.

**Figure 8 materials-15-00718-f008:**
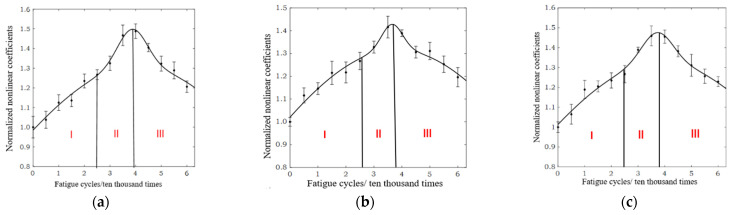
Three stages of the normalized ultrasonic nonlinear coefficients–fatigue cycles curve. (**a**) Specimen B1. (**b**) Specimen B2. (**c**) Specimen B3.

**Figure 9 materials-15-00718-f009:**
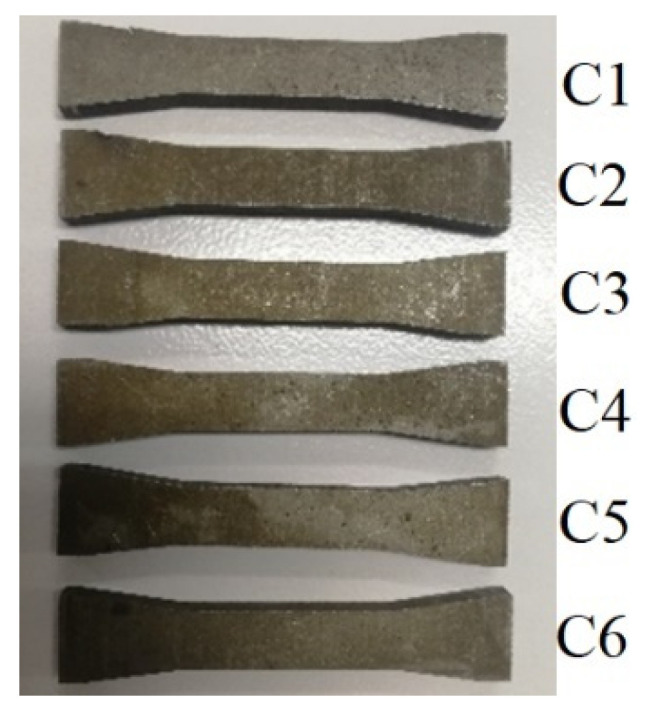
The SLM 316L stainless steel specimens (C1 to C6).

**Figure 10 materials-15-00718-f010:**
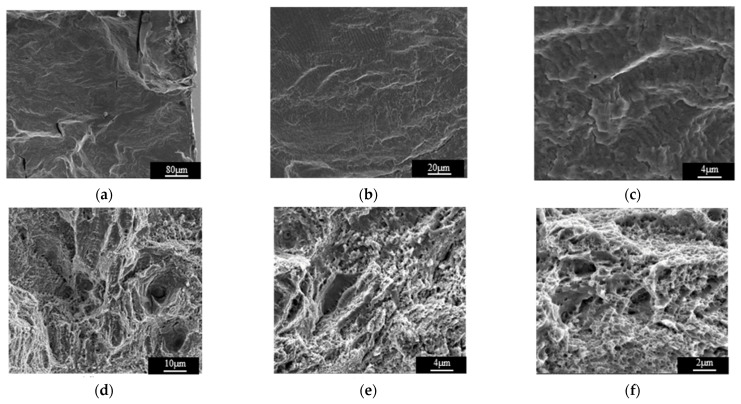
The SEM photographs of the fatigue fracture. (**a**) Fatigue source. (**b**) Low magnification of the fatigue expansion zone. (**c**) High magnification of the fatigue expansion zone. (**d**) Instantaneous fracture zone. (**e**) High magnification of the instantaneous fracture zone. (**f**) High magnification of the instantaneous fracture zone.

**Figure 11 materials-15-00718-f011:**
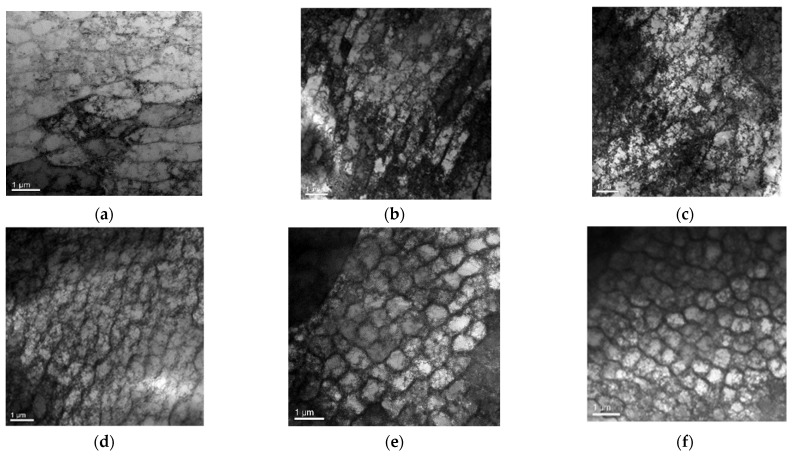
The TEM photographs of the SLM 316L stainless steel specimens subjected to different fatigue cycles. (**a**) Specimen C1 (10,000 times). (**b**) Specimen C2 (20,000 times). (**c**) Specimen C3 (30,000 times). (**d**) Specimen C4 (40,000 times). (**e**) Specimen C5 (50,000 times). (**f**) Specimen C6 (68,920 times).

**Figure 12 materials-15-00718-f012:**
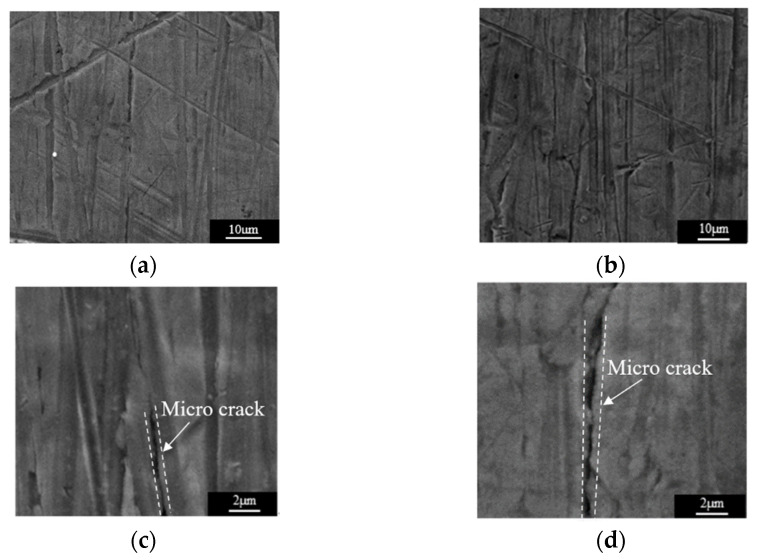
The SEM photographs of the SLM 316L stainless steel specimens subjected to different fatigue cycles. (**a**) Specimen C1 (10,000 times). (**b**) Specimen C2 (20,000 times). (**c**) Specimen C1 (30,000 times). (**d**) Specimen C2 (40,000 times).

**Figure 13 materials-15-00718-f013:**
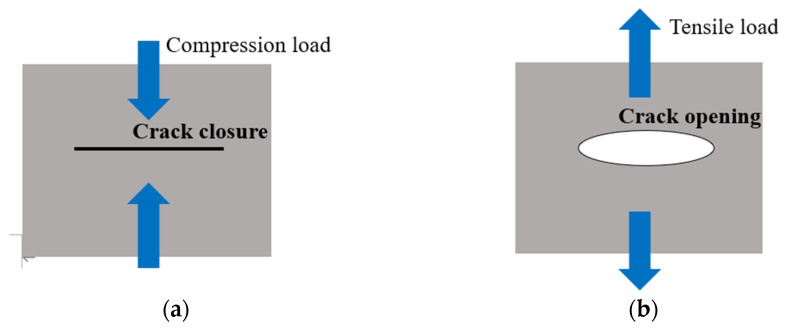
The breathing effect of the crack under a tensile and compressive load of the ultrasonic. (**a**) The crack is compressed and closed. (**b**) The crack is stretched and opened.

**Figure 14 materials-15-00718-f014:**
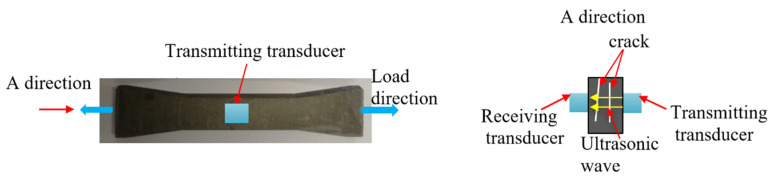
A schematic diagram of the ultrasonic propagation direction and crack direction.

**Figure 15 materials-15-00718-f015:**
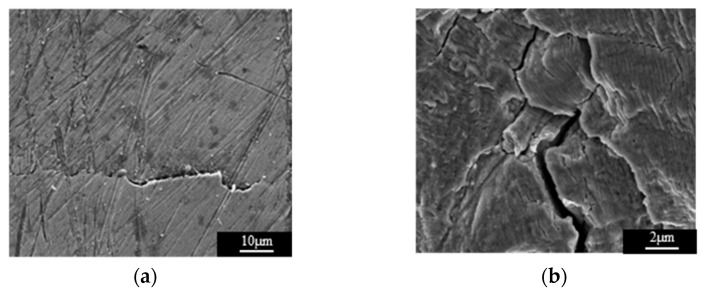
The SEM photographs of the SLM 316L stainless steel specimens subjected to different fatigue cycles. (**a**) Specimen C5 (50,000 times). (**b**) Specimen C6 (68,920 times).

**Table 1 materials-15-00718-t001:** Chemical composition of 316L stainless steel spherical powder (mass fraction, %).

Element	C	Cr	Ni	Mo	Si	Mn	O	P	Fe
Content	0.03	17.5	12.06	2.06	0.86	0.3	0.1	0.04	Bal.

**Table 2 materials-15-00718-t002:** Main processing parameters.

Laser Power (W)	Scanning Speed (mm/s)	Layer Thickness (μm)	Scanning Interval (mm)	Spot Diameter (μm)	Volume Fraction of Oxygen (%)
250	750	30	0.065	80	0.03

**Table 3 materials-15-00718-t003:** Fatigue loading parameters of the SLM 316L stainless steel specimens.

Specimen Number	C1	C2	C3	C4	C5	C6
Fatigue cycles (ten thousand times)	1	2	3	4	5	6.892
Maximum stress (MPa)	400	400	400	400	400	400
Stress ratio *r*	0.1	0.1	0.1	0.1	0.1	0.1
Loading frequency (Hz)	10	10	10	10	10	10

## Data Availability

The data presented in this study are available on request from the corresponding author.

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
