# Peer review of "The Characterization of Fatigue Damage of 316L Stainless Steel Parts Formed by Selective Laser Melting with Harmonic Generation Technique"

_materials, 2022, doi:10.3390/ma15030718_

Round 1

Reviewer 1 Report

The present work deals with fatigue damage monitoring of 316L stainless steel samples produced by SLM technology using nonlinear ultrasonic high-order harmonics detection method.

Although the topic of the article has moderately interesting, there are a few linguistic weaknesses in several places that must be remedied before publication.

Detailed comment related to the text:

Page 2: Equation numbers must be formatted correctly.

Page 5, line 153: Authors present in Figure 6 the received time domain signal and the frequency spectra of the received signal for a specimen subjected to 10000 fatigue cycles. Additional figures should be present for a different number of cycle (for instance 50000) to show second harmonic behaviour.

Page 5, line 165: This sub-section has the same designation as the section.

Page 7, line 216: Authors must give names to these samples shown in Figuire 9, to differentiate to samples previously analyse (B1 to B3).

Page 7, line 229: (E), should be replaced by (e).

Page 8, line 263: “fatigue” should be replaced by “Fatigue”.

Page 9, line 319: The word “was” should be deleted.

Page 9, line 323: The authors say: “The micro crack appears in the microstructure of the tested specimen”. The micro crack must be highlight in the figure to clarify the reader.

Page 10, line 351: This phrase must be reformulated, because the term “SEM images” were already present previously (Figure 12).

Page 10: Authors justify the behaviour of zone III of fatigue loading curves with the following sentence (line 353): “The distance between the crack surfaces is greater than the maximum displacement of the ultrasonic wave, the ultrasonic wave cannot directly pass through the crack, and no contact acoustic nonlinear effect is generated”. With the information presented in the paper, in referee opinion, is not possible to reach this conclusion. The authors should present detailed images with the orientation of the crack in relation to the ultrasonic propagation direction.

By other side, due to the load direction, predictably, the crack orientation will be perpendicular to the ultrasonic propagation direction, so the influence in the direct ultrasonic wave collected by the receiver transducer should be small.

Please comment this point since it is very important to justify the final evaluation of the normalized ultrasonic nonlinear coefficients-fatigue cycles curve.

Author Response

Dear Reviewer:

 Thank you very much for the time and effort that you have put into reviewing the previous version of the manuscript. Your suggestions have enabled me to improve my work greatly. Based on your comment and request, I have made extensive modification on the original manuscript. Here, I attached revised manuscript in the formats of PDF, for your approval. Appended to this letter is my point-by-point response to the comments raised by you.  The comments are reproduced and my responses are given directly afterward in a different color (red).

A revised manuscript with the correction sections red marked was attached as

the supplemental material and for easy check/editing purpose.  

 Should you have any questions, please contact me without hesitate. 

Kind regards

Yan XiaoLing

Reviewer 2 Report

Dear authors, 

Thank you for your interesting paper on damage detection in stainless steel produced by SLM. The paper is well written and offers interesting findings. I enjoyed reading your paper. However, I found several minor typos and linguistic errors. Before acceptance, I suggest the authors address the following aspects: 

•    The paper is perfectly understandable, but the grammar and spelling (e.g., use of articles, tenses, punctuation, singular / plural etc.) need more attention in some paragraphs. It is recommended to have a native English speaker check the manuscript. 

•    Abstract, line 9: It is unclear what those tests (nonlinear ultrasonic) are or what their goal is. Please add more information to the abstract. As a reader, I was wondering what it is about. 

•    line 24: You probably refer to cyclic or fatigue loading. Dynamic load only refers to the loading rate.

•    line 49: Eq. numbers should be on the right side according to the template.

•    line 51: You could add that sigma is time dependent.

•    line 88: Where all specimens produced together and can you report the interpass temperature, as it has a high effect on the mechanical properties? In addition, what is the build direction (e.g. vertical)?

•    line 97: There are a number of formatting problems.

•    line 191: Please revise the sentence. It is not fully clear what you want to express.

•    line 216: Please mention those test specimens in the beginning of the study. I was surprised to read that you know talk about different specimens. Were those new specimens manufactured with the others or separately?

•    line 226: What is a herringbone shape? I haven’t heard this term before.

•    line 319: Please revise the sentence. 

Author Response

Dear Reviewer:

 Thank you very much for the time and effort that you have put into reviewing the previous version of the manuscript. Your suggestions have enabled me to improve my work greatly. Based on your comment and request, I have made

extensive modification on the original manuscript. Here, I attached revised

manuscript in the formats of  PDF , for your approval. Appended to this letter is my point-by-point response to the comments raised by you.  The comments are reproduced and my responses are given directly afterward in a different color (red).

A revised manuscript with the correction sections red marked was attached as

 the supplemental material and for easy check/editing purpose.  

 Should you have any questions, please contact me without hesitate. 

Kind regards

Yan XiaoLing

Round 2

Reviewer 1 Report

In new Fig. 14 the cracks direction is perpendicular do wave propagation.

In Fig.13 the crack direction is perpendicular to load direction, so, due to the transducers position, the cracks direction will be parallel do ultrasonic wave propagation. This is contradictory with Fig. 14.

Please clarify this inconsistence.

Author Response

Dear Reviewer:

 Thank you very much for the time and effort that you have put into reviewing the previous version of the manuscript. Your suggestions have enabled me to improve my work greatly. Appended to this letter is my point-by-point response to the comments raised by you.  The comments are reproduced and my responses are given directly afterward in a different color (red).

 Should you have any questions, please contact me without hesitate. 

Kind regards

Yan XiaoLing
